# A High-Quality Reference Genome Assembly of *Prinsepia uniflora* (Rosaceae)

**DOI:** 10.3390/genes14112035

**Published:** 2023-11-02

**Authors:** Lei Zhang, Chaopan Zhang, Yajing An, Qiang Zhu, Mingcheng Wang

**Affiliations:** 1Key Laboratory of Ecological Protection of Agro-Pastoral Ecotones in the Yellow River Basin, National Ethnic Affairs Commission of the People’s Republic of China, College of Biological Science & Engineering, North Minzu University, Yinchuan 750021, China; zhangsanshi-0319@outlook.com (L.Z.); zcp15585866463@outlook.com (C.Z.); yajing_an@nmu.edu.cn (Y.A.); 2State Key Laboratory of Efficient Production of Forest Resources, Ningxia Forestry Institute, Yinchuan 750001, China; 12021140084@stu.nxu.edu.cn; 3Institute for Advanced Study, Chengdu University, No. 2025 Chengluo Road, Chengdu 610106, China; 4Engineering Research Center of Sichuan-Tibet Traditional Medicinal Plant, Chengdu 610106, China

**Keywords:** *Prinsepia uniflora*, medicinal plant, PacBio high-fidelity sequencing, chromosome-level genome assembly, genome annotation

## Abstract

This study introduces a meticulously constructed genome assembly at the chromosome level for the Rosaceae family species *Prinsepia uniflora*, a traditional Chinese medicinal herb. The final assembly encompasses 1272.71 megabases (Mb) distributed across 16 pseudochromosomes, boasting contig and super-scaffold N50 values of 2.77 and 79.32 Mb, respectively. Annotated within this genome is a substantial 875.99 Mb of repetitive sequences, with transposable elements occupying 777.28 Mb, constituting 61.07% of the entire genome. Our predictive efforts identified 49,261 protein-coding genes within the repeat-masked assembly, with 45,256 (91.87%) having functional annotations, 5127 (10.41%) demonstrating tandem duplication, and 2373 (4.82%) classified as transcription factor genes. Additionally, our investigation unveiled 3080 non-coding RNAs spanning 0.51 Mb of the genome sequences. According to our evolutionary study, *P. uniflora* underwent recent whole-genome duplication following its separation from *Prunus salicina*. The presented reference-level genome assembly and annotation for *P. uniflora* will significantly facilitate the in-depth exploration of genomic information pertaining to this species, offering substantial utility in comparative genomics and evolutionary analyses involving Rosaceae species.

## 1. Introduction

The Rosaceae family comprises over 3000 species, primarily distributed in temperate regions of the northern hemisphere [1]. Economically significant, this plant family boasts numerous renowned fruit-bearing plants, such as apple (*Malus*), pear (*Pyrus*), strawberry (*Fragaria*), hawthorn (*Crataegus*), and various *Prunus* species, including peach (*P. persica*), almond (*P. dulcis*), apricot (*P. armeniaca*), and plum (*P. salicina*). Additionally, it encompasses a range of ornamental plants with exquisite flowers, including the rose (*Rosa*), flowering cherry (*Cerasus*), crabapple (*Malus spectabilis*), and quince (*Cydonia oblonga*). Consequently, Rosaceae has garnered significant attention from plant breeders, who are dedicated to enhancing fruit quality and floral aesthetics [2,3,4].

The rapid advancement of genome sequencing technologies has substantially reduced the challenges associated with assembling high-quality genomes, offering promising prospects for precise plant breeding [5]. To date, genome sequencing efforts have successfully covered a substantial number of Rosaceae species, with most of the genome sequences accessible through the Genome Database for Rosaceae (GDR) [6]. As of September 2023, the GDR database houses a total of 128 genome assemblies, encompassing 11 genera and 59 Rosaceae species. These genomic datasets, complemented by other omics data, have greatly accelerated the breeding programs for Rosaceae plants.

*P. uniflora*, a deciduous shrub primarily found in Northwest China at altitudes ranging from 900 to 1100 m, has received notably less attention compared to other prominent members of the Rosaceae family. However, this plant possesses substantial medicinal value. The kernels of *P. uniflora*, referred to as ‘ruiren’ in China, have traditionally been used to cure eye conditions in traditional Chinese medicine [7]. Unfortunately, limited omics data have been published for *P. uniflora* thus far, impeding comprehensive genomic analyses of the biosynthetic pathways of its medicinal components. Recent phylogenetic investigations within the Rosaceae family have revealed that the genus *Prinsepia* is situated within the Exochordeae clade, closely related to Kerrieae and Sorbarieae [8,9]. The development of a high-quality genome assembly and annotation for *P. uniflora* will also prove invaluable for phylogenetic analyses, investigations into polyploidization events, and the study of karyotype evolution within Rosaceae.

In this study, we employed cutting-edge PacBio high-fidelity (HiFi) sequencing technology to assemble a high-quality genome of *P. uniflora*. Subsequently, we anchored this assembly onto chromosomes using high-throughput chromosome conformation capture (Hi-C) data. Leveraging this reference-level *P. uniflora* assembly, we conducted comprehensive annotation of repetitive sequences, protein-coding genes, and non-coding RNAs (ncRNAs) utilizing diverse bioinformatics methodologies. The presentation of this high-quality *P. uniflora* genome assembly and its associated annotation will provide the scientific community with a valuable genomic asset, facilitating the in-depth exploration of *P. uniflora*’s genomic information and supporting genetic and genomic inquiries within the Rosaceae family.

## 2. Materials and Methods

### 2.1. Sample Collection

Leaf specimens for genome sequencing were obtained from a 2-year-old individual *P. uniflora* plant located in the Yinchuan Botanical Garden, Ningxia Province, Northwestern China. Fresh samples of leaves, stems, flowers, and roots were harvested from the same plant in its flowering period. Subsequently, all plant materials were immediately flash-frozen in liquid nitrogen before DNA or RNA extraction.

### 2.2. Genome Survey

We performed the extraction of total genomic DNA utilizing the cetyl trimethylammonium bromide (CTAB) method. The NEBNext Ultra II DNA Library Prep Kit (New England Biolabs, MA, USA) was used to create paired-end Illumina ReSeq libraries with an average insert size of 400 bp. These libraries were then subjected to sequencing on an Illumina NovaSeq 6000 platform (Illumina Inc., San Diego, CA, USA). Following this, we estimated the haploid genome size and heterozygosity rate of *P. uniflora* by analyzing the *k*-mer (*k* = 19) distribution frequency of Illumina sequencing reads, employing Jellyfish v2.2.9 [10].

### 2.3. PacBio HiFi Sequencing and Assembly

A modified CTAB approach was used to extract high-molecular-weight DNA. We generated HiFi reads using the PacBio Sequel II platform (Pacific Biosciences, Menlo Park, CA, USA) in circular consensus sequencing (CCS) mode, adhering to the PacBio 15 kb protocol to simplify PacBio HiFi sequencing. The resulting HiFi reads underwent preprocessing via the CCS analysis workflow within SMRT Link v8.0 (PacBio) and were subsequently assembled into a contig-level assembly employing hifiasm v0.14 [11]. Using Purge Haplotigs v1.1.1, possible duplicate haplotypes were found and eliminated from the original assembly. [12]. Additionally, we mapped Illumina reads back to the polished assembly using BWA v0.7.17 [13]. Pseudo-contigs with exceptionally low coverage depth (<5×) and high GC content (>50%) were excluded.

### 2.4. Hi-C Sequencing and Scaffolding Analysis

Hi-C libraries were created via chromatin extraction and digestion, DNA ligation, and purification, all in accordance with a predetermined protocol [14]. Then, these libraries were sequenced using an Illumina NovaSeq 6000 platform. Using Juicer v1.8.8 [15], Hi-C paired-end reads were aligned back to the contigs, preserving uniquely mapped Hi-C reads. Finally, the contigs were anchored into pseudochromosomes using the 3D-DNA program [16].

### 2.5. RNA Sequencing

For RNA sequencing (RNA-seq), we isolated total RNA from fresh leaf, stem, flower, and root tissues of *P. uniflora* using TRIzol reagent. After eliminating residual DNA, RNA-seq libraries were generated using the NEBNext Ultra II RNA Library Prep Kit and sequenced using an Illumina NovaSeq 6000 platform. The resulting RNA-seq reads were subjected to filtration using Trimmomatic v0.36 [17] prior to transcriptome-based gene prediction.

### 2.6. Genome Assessment

We assessed the completeness of both the genome assembly and the protein-coding gene set by calculating the benchmarking universal single-copy orthologs (BUSCO) completeness score. This analysis employed BUSCO v3.0.2 [18] and utilized the Embryophyta odb10 dataset. Furthermore, we computed long terminal repeat (LTR) assembly index (LAI) scores using LTR_retriever v2.8 [19] with a sliding window size of 3 Mb.

### 2.7. Repeat Annotation

We generated a de novo repeat library based on the genome assembly using RepeatModeler v2.0.1 [20]. Later on, this library was merged with the green plant repeat library from the Repbase database version 22.11 [21]. Finally, we conducted homology-based detection of repetitive elements using RepeatMasker v4.1.0 [22].

Additionally, we predicted full-length LTR retrotransposons (LTR-RTs) using previously established methods [23]. With the aid of LTR_Finder v1.06 [24] and LTRharvest v1.5.10 [25], candidate intact LTR-RTs were located. LTR_retriever was then used to integrate the final forecasts.

### 2.8. Gene Prediction and Functional Annotation

Based on the repeat-masked assembly, we used a combination of homology-based searches, de novo prediction, and transcript alignment to annotate protein-coding genes. First, using Program to Assemble Spliced Alignment (PASA) v2.3.3 [26], transcripts were assembled using RNA-seq data, and the alignment findings were used to estimate gene architectures.

Secondly, four software applications, namely, GlimmerHMM v3.04 [27], SNAP v2013.11.29 [28], GeneMark-ES v4.33 [29], and Augustus v3.2.3 [30], were employed for the de novo prediction of genes.

Thirdly, we aligned protein sequences from various related species, including *Prunus avium* [31], *P. mume* [32], *P. persica* [33], *P. dulcis* [34], *P. armeniaca* [35], and *Rosa chinensis* [36], to the genome assembly using TBLASTN v2.2.31 [37]. Gene structures were then predicted based on the alignments using GeneWise v2.4.1 [38].

Finally, EVidenceModeler v1.1.1 was used to combine all predicted gene models into a final protein-coding gene set [39]. Tandemly duplicated genes were identified based on their proximity (within 10 consecutive genes and 100 kb) and similarity (identity ≥ 50% and coverage ≥ 70%), as previously described by Wang et al. (2022) [40]. The plant transcription factor database (PlantTFDB) v5.0 was used to predict TF genes [41].

Aligning the protein sequences of the protein-coding genes with Swiss-Prot [42], eggNOG [43], and the NCBI non-redundant protein databases (NR) allowed for functional annotation. InterProScan v5.35 [44] was used to annotate protein domains and motifs, and the matching InterPro entries were used to assign gene ontology (GO) terms. The KEGG Automatic Annotation Server was used to annotate gene pathways [45]. Additionally, ncRNAs were annotated using tRNAscan-SE v2.0 [46], BLASTN v2.2.31, and Infernal v1.1.2 [47], following methods previously described [40].

### 2.9. Evolutionary Analysis

The protein sequences of *P. uniflora* and eight other sequenced Rosaceae species, *Malus domestica* (HFTH1 v1.0), *Pyrus communis* (Bartlett DH v2.0), *Gillenia trifoliata* (v1.0), *Eriobotrya japonica* (v1.0), *Crataegus pinnatifida* (v1.0), *P. salicina* (Sanyueli v2.0), *Potentilla anserina* (v1.0), and *Rosa rugosa* (v1.0), were used for the phylogenetic analysis. OrthoFinder v2.3.11 [48] was used to identify orthologous and paralogous gene groupings among the nine species, with the options “-S diamond -M msa.”. Protein sequences were aligned using MAFFT-LINSI v7.313 for each single-copy orthogroup [49]. RAxML v8.2.11 [50] was used to create a maximum likelihood species tree based on the concatenated alignments of all single-copy orthogroups, with the settings “-f a -m PROTGAMMAILGX -N 200.”.

MCMCTREE in PAML v4.9e was used to estimate the divergence times among the nine species [51]. The fossil calibration points were the *R. rugosa*–*M. domestica* divergence (49.2–77.1 million years ago, Mya) and the *P. salicina*–*M. domestica* divergence (34.4–67.2 Mya) that were taken from the TimeTree database (http://www.timetree.org, Accessed on 9 August 2023).

Whole-genome duplication (WGD) analysis was carried out based on all-against-all pairwise comparisons of protein sequences using DIAMOND v0.9.22 [52]. MCScanX v1.1 was used to identify syntenic gene pairings both within and between genomes [53]. Each gene pair’s synonymous substitution rate (*K*s) was determined using the MCScanX script “add_ka_and_ks_to_collinearity.pl.”. By examining the *K*s distributions of orthologous and paralogous gene pairs within and between species, the occurrence of WGD events was investigated.

## 3. Results and Discussion

### 3.1. Genome Sequencing and Assembly

Prior to assembly, we generated a substantial 91.20 gigabases (Gb) of Illumina paired-end reads for *k*-mer analysis (Appendix A). Subsequently, we estimated the haploid genome size of *P. uniflora* to be 1.33 Gb, with a heterozygosity rate of 0.41% (Appendix A). Given the low heterozygosity of the *P. uniflora* genome, we chose to assemble a collapsed assembly instead of haplotype-resolved assemblies in this study. We then generated a total of 26.77 Gb of PacBio HiFi reads (number of reads = 1,639,479; N50 = 16.14 kb) for the de novo assembly of the *P. uniflora* genome (Appendix A). This effort yielded 3672 contigs covering 1.68 Gb with an N50 of 2.2 Mb (Appendix A). After excluding potential duplicate haplotypes and pseudo-contigs, we retained 820 contigs, totaling 1.27 Gb, for subsequent Hi-C scaffolding analysis (Appendix A). Ultimately, 99.84% of the entire genome assembly was anchored onto 16 pseudochromosomes (Figure 1a), utilizing 106.49 Gb of Hi-C paired-end reads. The pseudochromosomes varied in length, ranging from 67.28 Mb to 96.32 Mb (Appendix A). This pseudochromosome number aligns with findings from a previous karyotype study of the Rosaceae family [54]. We observed an overall GC content of 41.68% for the final assembly (Figure 1b), with contig and super-scaffold N50 values of 2.77 and 79.32 Mb, respectively (Table 1 and Appendix A). The BUSCO completeness score for the final assembly reached 97.03%, encompassing 51.36% single-copy and 45.66% duplicated BUSCOs (Appendix A). Additionally, we noted an overall LAI score of 22.86 (standard deviation = 5.15) for the entire genome, indicating a high level of completeness (Appendix A).

### 3.2. Repeat Annotation

Within the final assembly, we identified a total of 875.99 Mb of repetitive sequences, constituting 68.83% of the *P. uniflora* genome (Appendix A). Transposable elements (TEs) constituted the majority of repeat sequences, spanning 777.28 Mb or 61.07% of the entire genome (Figure 1b). This was followed by 113.34 Mb of unclassified repeats and a smaller proportion of other repeat classes, including satellites (2.72 Mb), simple repeats (1.11 Mb), and low-complexity repeats (205 bp). LTR-RTs accounted for 94.64% (735.58 Mb) of the TE sequences, with a *Gypsy*/*Copia* ratio of 2.37. Furthermore, we identified 13,071 full-length LTR-RTs with a cumulative length of 117.78 Mb within the *P. uniflora* genome. This included 6295 *Gypsy* elements, 2039 *Copia* elements, and 4737 unclassified LTRs (Appendix A).

### 3.3. Gene Prediction and Functional Annotation

We generated a comprehensive dataset of 27.95 Gb of RNA-seq data for transcriptome-based gene prediction (Appendix A), which resulted in the identification of 69,894 gene models. These gene models were amalgamated with predictions from homology-based and de novo approaches (Appendix A), culminating in a final consensus gene set encompassing 49,261 protein-coding genes, achieving a BUSCO completeness score of 97.34% (Appendix A). The average transcript length of the estimated genes was 3816 bp, the average coding sequence (CDS) size was 1441 bp, each gene had an average of 5.3 exons, and the average intron length was 557 bp (Table 1 and Appendix A). The 16 pseudochromosomes contained 99.77% of the genes (49,148), resulting in an overall gene density of 39.7 genes per Mb (Appendix A). Among these protein-coding genes, 2373 (4.82%) were identified as transcription factor (TF) genes, with the bHLH, MYB, and ERF families being the three largest TF families (Appendix A). A total of 45,256 (91.87%) genes were assigned functional annotations from public databases, and 14,696 (29.83%) genes were attributed to GO terms (Appendix A). Additionally, we identified 5127 (10.41%) tandemly duplicated genes distributed across 1952 arrays within the *P. uniflora* genome (Appendix A). These genes were significantly enriched in processes related to “obsolete oxidation-reduction”, “metabolism”, “defense response”, and “transmembrane transport” (Appendix A). Furthermore, we annotated a total of 3080 ncRNAs spanning 0.51 Mb of genome sequences, encompassing 957 transfer RNAs (tRNAs), 181 microRNAs (miRNAs), 1216 ribosomal RNAs (rRNAs), and small nuclear RNAs (snRNAs) (Appendix A). Collectively, these annotated gene sets will substantially facilitate further genomic and genetic studies of *P. uniflora*.

### 3.4. Evolutionary History of P. uniflora

*P. uniflora* and eight other Rosaceae species were found to have 118 single-copy orthogroups according to our OrthoFinder investigation. Based on these single-copy orthogroups, a well-supported species tree was recovered, showing that *P. uniflora* and *P. salicina* were clustered together with nearly full (97%) bootstrap support (Figure 2a). The divergence time between *P. uniflora* and *P. salicina* was estimated to be around 37.6 Mya, which was slightly later than the divergence between their common ancestor and the clade formed by Maleae and Gillenieae (44.9 Mya). We observed a major peak around 0.29 in the *K*s distribution of orthologues between *P. uniflora* and *P. salicina*, and a younger peak around 0.12 in the paralog analysis of *P. uniflora* (Figure 2b). Therefore, we speculated that an independent WGD event had occurred in the *P. uniflora* genome after its split from *P. salicina*, which may have contributed significantly to its larger haploid genome size (1.27 Gb vs. 284 Mb) and larger number of protein-coding genes (49,261 vs. 27,481) compared to *P. salicina*. According to the Plant DNA C-values Database, another Exochordeae species—*Exochorda giraldii*—has eight haploid chromosomes with a haploid genome size of smaller than 600 Mb. Therefore, this recent WGD event likely occurred in *P. uniflora* after its split from *E. giraldii*.

Notably, there are some differences between our species tree and the nuclear phylogeny presented by Xiang et al. (2017) [8]. In the previous phylogeny, *P. uniflora* was more closely related to the clade formed by Maleae and Gillenieae. However, our nuclear genome-based phylogeny seems more reasonable because both *P. uniflora* and *P. salicina* had chromosome numbers that are multiples of 8, while all Maleae species had 17 chromosomes and Gillenieae had 9 (Figure 2a). In the future, more high-quality genome sequences of Rosaceae species will help us further elucidate the phylogenetic relationships and evolutionary history of Rosaceae.

## 4. Conclusions

In this study, we successfully assembled a chromosome-level genome of *P. uniflora* using advanced PacBio HiFi sequencing and Hi-C technologies. The resultant *P. uniflora* assembly exhibits a commendable level of continuity and completeness, albeit with a notable presence of repetitive elements. Subsequent gene predictions conducted on this high-quality genome assembly yielded a substantial set of 49,261 protein-coding genes and 3080 non-coding RNAs (ncRNAs). Significantly, a substantial portion of these protein-coding genes underwent functional annotation, offering valuable insights for forthcoming functional genomic inquiries pertaining to *P. uniflora*. Furthermore, *P. uniflora* and *P. salicina* clustered together according to comparative genomic analysis, and the *P. uniflora* genome had recently undergone a WGD event.

The high-quality genome assembly and meticulous annotation of *P. uniflora* presented herein are poised to expedite comprehensive genome-wide investigations concerning the biosynthesis of medicinal constituents in this traditional Chinese medicinal plant. Furthermore, it serves as a valuable addition to the ongoing comparative genomics endeavors within the Rosaceae family.

## 5. Significance

This study marks the inaugural genome assembly and annotation of the *Prinsepia* species, thereby endowing the scientific community with a valuable genomic reservoir for advancing research endeavors related to *P. uniflora*. Moreover, it serves as a valuable contribution to the broader field of comparative genomics within the Rosaceae family.

## Figures and Tables

**Figure 1 genes-14-02035-f001:**
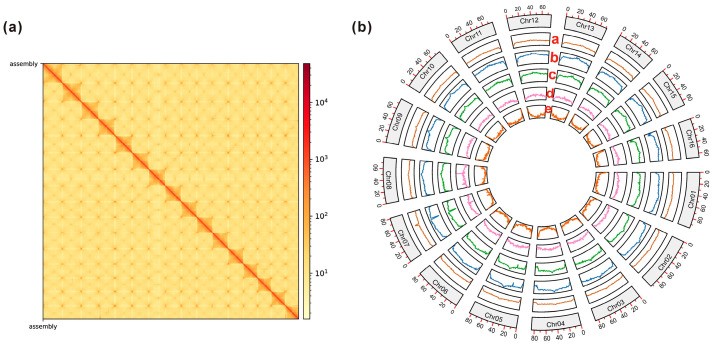
Characteristics of the *P. uniflora* genome assembly. (**a**) Heatmap displaying the Hi-C interactions among the 16 pseudochromosomes of the *P. uniflora* genome. (**b**) *P. uniflora*’s genomic features are displayed in non-overlapping windows of 1 Mb, and the tracks show the following: (a) GC content; (b) repeat density; (c) TE density; (d) density of unclassified repeats; and (e) gene density.

**Figure 2 genes-14-02035-f002:**
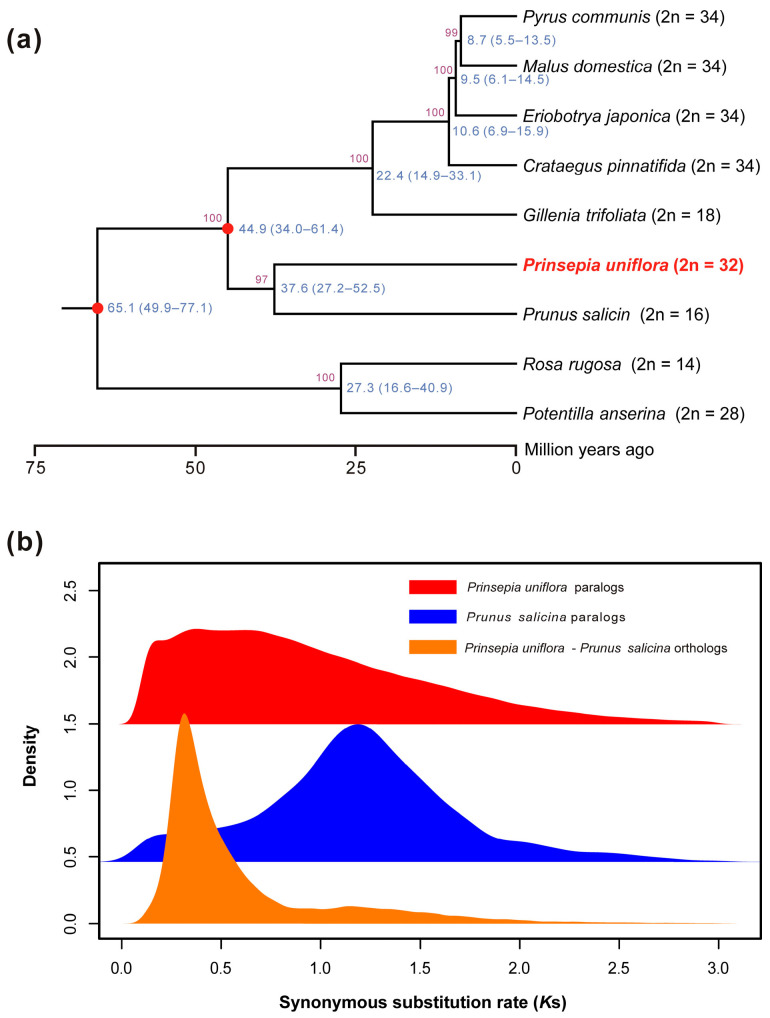
Phylogenetic and evolutionary analysis of the *P. uniflora* genome. (**a**) A species tree comprising eight other Rosaceae species and *P. uniflora*, based on 118 single-copy orthogroups. Divergence times with 95% confidence intervals are indicated in blue, while bootstrap values are indicated in purple. Red dots represent the fossil calibration points. Chromosome numbers are positioned to the right of species names. (**b**) *K*s distributions for the paralogs and orthologs found in *P. uniflora* and *Prunus salicina*’s entire genomes.

**Table 1 genes-14-02035-t001:** Overall statistics regarding the annotation and assembly of the *P. uniflora* genome.

**Assembly**	
Haploid genome size (estimated via *k*-mer analysis) (Mb)	1329.36
GC content (%)	41.68
Total length (Mb)	1272.71
Pseudochromosome number	16
Pseudochromosome length (Mb)	1270.71
Gap size (bp)	401,500
Super-scaffold N50 (Mb)	79.32
Contig N50 (Mb)	2.77
Max contig length (Mb)	16.35
BUSCO score (%)	97.03
**Annotation**	
Repeat content (%)	68.83
Number of protein-coding genes	49,261
Average length of transcript (bp)	3816
Average length of coding sequence (bp)	1441
Average length of exons (bp)	274
Average length of introns (bp)	557
Average exons per gene	5.3
Functionally annotated genes	45,256
Tandemly duplicated genes	5127
Transcription factor genes	2373
Non-coding RNAs	3080
BUSCO score (%)	97.34

## Data Availability

Under BioProject PRJNA1017129, the *P. uniflora* assembly and all sequencing data have been submitted to the NCBI. Sequence Read Archive (SRA) accession numbers SRR26058176, SRR26058177, SRR26058178, and SRR26058179 can be used to obtain the RNA-seq data. SRR26448902, SRR26457063, and SRR26426586 are the SRA accession numbers for whole-genome Illumina, HiFi, and Hi-C data. Also accessible on FigShare at https://doi.org/10.6084/m9.figshare.24137688.v1 (Accessed on 20 September 2023) are the genome assembly and annotations.

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
