# Peer review of "A High-Quality Reference Genome Assembly of Prinsepia uniflora (Rosaceae)"

_genes, 2023, doi:10.3390/genes14112035_

Round 1

Reviewer 1 Report

Comments and Suggestions for Authors

In the brief Report entitled "A high-quality reference genome assembly of Prinsepia uniflora (Rosaceae)", the authors employed cutting-edge PacBio high-fidelity (HiFi) sequencing technology to assemble a high-quality genome of Prinsepia uniflora. They anchored the assembly onto chromosomes using high-throughput chromosome conformation capture (Hi-C) data. They conducted comprehensive annotation of repetitive sequences, protein-coding genes, and non-coding RNAs (ncRNAs) utilizing diverse bioinformatics methodologies. The authors concluded that the presentation of this high-quality P. uniflora genome assembly and associated annotations will provide the scientific community with valuable genomic assets, facilitating in-depth exploration of genomic information of P. uniflora and supporting genetic and genomic inquiries within the Rosaceae family. This study is interesting and clear with a valid and better selection of samples. The study included well-presented data and analysis, and tables and figures are clarified. However, minor revisions are needed as follows:

- Keywords: Please do not use the same words in the title to repeat them in the keywords as much as possible.

- In general, please check the manuscript for grammatical errors and typos.

Comments on the Quality of English Language

Minor editing of the English language is required.

Author Response

In the brief Report entitled "A high-quality reference genome assembly of Prinsepia uniflora (Rosaceae)", the authors employed cutting-edge PacBio high-fidelity (HiFi) sequencing technology to assemble a high-quality genome of Prinsepia uniflora. They anchored the assembly onto chromosomes using high-throughput chromosome conformation capture (Hi-C) data. They conducted comprehensive annotation of repetitive sequences, protein-coding genes, and non-coding RNAs (ncRNAs) utilizing diverse bioinformatics methodologies. The authors concluded that the presentation of this high-quality P. uniflora genome assembly and associated annotations will provide the scientific community with valuable genomic assets, facilitating in-depth exploration of genomic information of P. uniflora and supporting genetic and genomic inquiries within the Rosaceae family. This study is interesting and clear with a valid and better selection of samples. The study included well-presented data and analysis, and tables and figures are clarified.

Response: Thanks for your positive comments on our manuscript.

However, minor revisions are needed as follows:

- Keywords: Please do not use the same words in the title to repeat them in the keywords as much as possible.

Response: After replacing Rosaceae with medicinal plant, there was only one word (Prinsepia uniflora) appeared in both the keywords and title.

- In general, please check the manuscript for grammatical errors and typos.

Response: Our manuscript has been edited by a native English-speaking editor of MogoEdit. We have carefully checked the grammatical errors and typos before the 2nd round of submission.

Reviewer 2 Report

Comments and Suggestions for Authors

Assembling a genome is a very labor-intensive, painstaking experiment. The work makes a major contribution to the genome database. It should also be noted the use of new high technologies. It is possible, as a small comment, to point out that almost all illustrations and tables have been moved to the Supplementary section and are practically not discussed and there are no comparisons with the genomes of other representatives of the Rosaceae family

Author Response

Assembling a genome is a very labor-intensive, painstaking experiment. The work makes a major contribution to the genome database. It should also be noted the use of new high technologies. It is possible, as a small comment, to point out that almost all illustrations and tables have been moved to the Supplementary section and are practically not discussed and there are no comparisons with the genomes of other representatives of the Rosaceae family.

Response: Thanks for your positive comments and useful suggestions. Because our manuscript type is Brief Report, we have only included one figure and one table in the main text. As suggested by you and reviewer 3, we have added the results of comparative genomics analysis into the manuscript.

Reviewer 3 Report

Comments and Suggestions for Authors

Zhang et al assembled and annotated a reference genome for Prinsepia uniflora. Types of annotations included repetitive elements, transposable elements, protein-coding genes, non-coding RNAs. The authors presented the findings in a concise, well-written manner. More detail could be given to materials and methods as well as providing context for how this assembly compares to others in the family. Other major concerns are with the assembly method, genome size and chromosome number (raising whole genome duplication/ploidy possibility) and data availability.

Major:

Hifiasm produces haplotype resolved assemblies as well as a collapsed assembly. Why aren’t the haplotype resolved assemblies reported? The explanation for not addressing them could be included to clarify. It is especially efficient at this when Hi-C data is included during assembly, why wasn’t it included (if it wasn’t?)? It also purges duplicates internally, so why use purge haplotigs software?

The genome that was made available on Figshare is only 1.19Gigabyes but the stated genome size is 1.27Gigabases?

This genome is quite an outlier for Rosaceae, Prunus has a base chromosome level of 8; Malus and Pyrus are 17 but only after a confirmed recent WGD. In the tribe Exochrodeae, the Kew gardens C-value database does not have an entry for Prinsepia but does have one for Exochorda giraldii, with 8 haploid chromosomes and a 588Mb haploid genome. This is further brought into question by the high number of genes (49k) and the busco duplication rate of 50%. It seems likely that this species has gone through recent whole genome duplication.  Whole genome duplication analysis needs to be completed, and the discussion should address the comparison of this genome with the rest of the family.

Data Availability:

Genome, annotation, Illumina whole genome, Illumina Hi-C and PacBio reads are not available and need to be submitted to NCBI. (RNASeq are available).

Minor:

Methods need additional detail:

Line 73: What rough developmental stages were the tissues collected at?

Line 78: What library prep kit was used? Please include manufacturer and city.

Line 85: How did you extract HMW DNA?

Line 100: Did you do any manual manipulation of the assembly with Juicebox? Please detail changes if so.

Lines 102-103: What method was used to extract RNA? What library prep kit was used? What tissues were used?

Line 165: Make Figure 1B larger and a-e labels stand out more. Perhaps consider adding a background color to those text boxes.

Please double check References. Reference #48 is not cited within manuscript.

Author Response

Zhang et al assembled and annotated a reference genome for Prinsepia uniflora. Types of annotations included repetitive elements, transposable elements, protein-coding genes, non-coding RNAs. The authors presented the findings in a concise, well-written manner.

Response: Thanks for your positive comments on our submitted work.

More detail could be given to materials and methods as well as providing context for how this assembly compares to others in the family. Other major concerns are with the assembly method, genome size and chromosome number (raising whole genome duplication/ploidy possibility) and data availability.

Major:

Hifiasm produces haplotype resolved assemblies as well as a collapsed assembly. Why aren’t the haplotype resolved assemblies reported? The explanation for not addressing them could be included to clarify. It is especially efficient at this when Hi-C data is included during assembly, why wasn’t it included (if it wasn’t?)? It also purges duplicates internally, so why use purge haplotigs software?

Response: Hifiasm is a powerful tool that can produce haplotype resolved assemblies, especially for genomes that are highly heterozygous. However, we observed a low heterozygosity rate (0.41%) for the Prinsepia uniflora genome based on k-mer frequency analysis. Thus, it is not necessary to generate haplotype resolved assemblies in this study. We have added the results and explanation into the main text, and also updated Figure S1. Since we decided to generate a collapsed assembly, Hi-C data was not included during the de novo assembly.

Hifiasm can purge duplicates internally. However, it cannot detect and remove all duplicate haplotypes with some genomes using default parameters. Thus, we used purge haplotigs software to remove duplicate haplotypes which cannot be detected by Hifiasm.

The genome that was made available on Figshare is only 1.19 Gigabyes but the stated genome size is 1.27 Gigabases?

Response: In windows system, 1 Gigabyes = 1,024 Mb, and 1 Mb = 1,024 kb. However, all genome papers calculate genome size in the form of 1 Gb = 1,000 Mb, 1 Mb = 1,000 kb. As a result, there are some subtle differences in the amount of data displayed.

This genome is quite an outlier for Rosaceae, Prunus has a base chromosome level of 8; Malus and Pyrus are 17 but only after a confirmed recent WGD. In the tribe Exochrodeae, the Kew gardens C-value database does not have an entry for Prinsepia but does have one for Exochorda giraldii, with 8 haploid chromosomes and a 588Mb haploid genome. This is further brought into question by the high number of genes (49k) and the busco duplication rate of 50%. It seems likely that this species has gone through recent whole genome duplication. Whole genome duplication analysis needs to be completed, and the discussion should address the comparison of this genome with the rest of the family.

Response: Thanks for your critical advice. As suggested by you and reviewer 3, we have added the results of comparative genomics analysis into the manuscript.

Data Availability:

Genome, annotation, Illumina whole genome, Illumina Hi-C and PacBio reads are not available and need to be submitted to NCBI. (RNA-Seq are available).

Response: We have deposited the genome assembly, Illumina whole genome, Illumina Hi-C and PacBio reads to NCBI under the BioProject PRJNA1017129. We have contacted NCBI by e-mail, and they will release the annotation files after they finish the verification process. We have updated the Data Availability Statement.

Minor:

Methods need additional detail:

Line 73: What rough developmental stages were the tissues collected at?

Response: The tissues were collected at the flowering period of a 2-year-old plant. We have added the description into the main text.

Line 78: What library prep kit was used? Please include manufacturer and city.

Response: We used NEBNext Ultra II DNA Library Prep Kit (New England Biolabs, MA, USA). We have added the description into the main text.

Line 85: How did you extract HMW DNA?

Response: High molecular weight DNA was extracted using a modified CTAB method. We have added the description into the main text.

Line 100: Did you do any manual manipulation of the assembly with Juicebox? Please detail changes if so.

Response: No obvious scaffolding errors were detected. Thus, we didnt do any manual manipulation of the assembly with Juicebox.

Lines 102-103: What method was used to extract RNA? What library prep kit was used? What tissues were used?

Response: We isolated total RNA from fresh leaf, stem, flower, and root tissues of P. uniflora using TRIzol reagent. RNA-seq libraries were generated using the NEBNext Ultra II RNA Library Prep Kit. We have added the description into the main text.

Line 165: Make Figure 1B larger and a-e labels stand out more. Perhaps consider adding a background color to those text boxes.

Response: Thanks for your suggestion. We have made Figure 1B larger to make it more clear to read. We have also made a-e labels bold, changed their positions, and changed their color into red.

Please double check References. Reference #48 is not cited within manuscript.

Response: We have added several references in the “Materials and Methods” and double checked the references in the main text.